# Phosphatidic Acid Reverses Obesity Induced by a High-Fat, High-Sugar Diet at the Transcriptional Level

**DOI:** 10.3390/genes16091112

**Published:** 2025-09-19

**Authors:** Hao Xie, Qian Cheng, Xingyi Tian, Yanlin Liao

**Affiliations:** 1School of Life Science and Technology, the Key Laboratory of Developmental Genes and Human Disease, Southeast University, 2 Dongda Road, Nanjing 210031, China; tianxingyi0510@163.com (X.T.); 18137311278@163.com (Y.L.); 2Department of Pathogen Biology Microbiology Division, Key Laboratory of Pathogen of Jiangsu Province, Nanjing Medical University, Nanjing 211166, China; chengqian@njmu.edu.cn

**Keywords:** phosphatidic acid, high-fat, high-sugar diet, obesity, transcriptome, PPAR

## Abstract

**Background:** Obesity poses a significant threat to human health and is commonly caused by excessive dietary intake. Phosphatidic acid (PA) is one of the simplest diacylglycerol phospholipids, serving as a crucial precursor for the synthesis of triglycerides and other complex phospholipids. PA is also an important intermediate product in the process of fat digestion and absorption. Studies have shown that PA has muscle-building and fat-reducing effects, but it is currently unclear whether it can combat obesity induced by a high-fat, high-sugar diet (HFD). **Methods:** Using a model of obesity induced by a high-fat high-sugar diet, we found that the addition of PA to food could reverse HFD-induced obesity. **Results:** Addition of PA to food can reverse obesity induced by a high-fat diet. Transcriptomic analysis results indicate that this reversal also takes place at the molecular level. Further analysis suggests that PA may regulate fat metabolism by reversing the PPAR signaling pathway. **Conclusions:** Our study provides molecular evidence for the use of PA as an effective additive in weight-loss food products.

## 1. Introduction

Obesity is a chronic disease characterized by the excessive accumulation of fat that has become an issue worldwide [1,2]. Although the exact etiology of obesity is not fully understood, unhealthy lifestyles (including poor dietary habits and lack of proper exercise), genetic factors, and obesogenic environments are significant contributors [3]. Fat is an essential bodily component [4]; in addition to being a major constituent of cell membranes, lipid molecules also participate in signal pathway regulation and energy storage and influence individual growth, development, and normal physiological functions [5,6]. Adipose tissue serves as a crucial storage site for triglycerides and cholesterol esters, playing a pivotal role in regulating lipid metabolism [7]. Excessive fat accumulation in adipose tissue leads to obesity [8]. High-calorie diets and a lack of physical activity are key factors that can contribute to excessive fat accumulation [7]. While extensive research has been conducted to investigate the molecular mechanisms underlying fat production and regulation and to develop pharmacological and non-pharmacological treatment strategies to combat obesity [9,10,11,12,13], safe and effective methods to counteract obesity are still lacking.

Maintaining lipid homeostasis is crucial for cellular function, as imbalances are closely associated with various metabolic disorders, including obesity, diabetes, and cancer [14]. In eukaryotic cells, lipid metabolism is intricately regulated by multilayered gene networks, which can be classified based on their functions into nuclear receptors [15], transcription factors [16], enzymes, and membrane contact site-related genes [17]. These genes collectively coordinate the synthesis, storage, and breakdown of intracellular lipids to maintain lipid homeostasis. Phosphatidic acid (PA), a pivotal molecular node in lipid metabolism, plays a central role in this network as both a precursor for various phospholipids and a crucial signaling molecule. The synthesis of PA primarily occurs through two pathways: the acyltransferase pathway and the phospholipase D pathway. In the acyltransferase pathway, GPAT and AGPAT sequentially catalyze glycerol-3-phosphate acylation, ultimately producing PA [18,19]. This process mainly takes place at the outer mitochondrial membrane and endoplasmic reticulum. In the phospholipase D pathway, typically activated during signal transduction processes, PLD hydrolyzes phosphatidylcholine (PC) and other phospholipids to generate PA [20]. In the metabolic fate of PA, two primary processes occur: first, dephosphorylation by PAP/Lipin enzymes to generate DAG, a crucial branch point for TAG and phospholipid synthesis [21]; second, conversion by CDP-DAG synthase (CDS) to CDP-DAG, leading to the synthesis of phosphatidylglycerol (PG) and cardiolipin (CL), among other acidic phospholipids [22]. This metabolic network situates PA at the core of lipid metabolism, directly influencing the composition and balance of various lipids. Phosphatidic acid (PA) plays a central regulatory role in maintaining lipid homeostasis, primarily by modulating the synthesis and breakdown of triglycerides (TAGs) to impact the formation and dynamic changes of lipid droplets, thereby maintaining energy balance. The main pathway for TAG synthesis is the Kennedy pathway, in which PA occupies a central role PA is dephosphorylated by PAP/Lipin enzymes to form DAG, which then undergoes acylation by DGAT enzymes with fatty acyl-CoA to produce TAGs. The flux through this pathway largely determines the intracellular accumulation levels of TAGs [23,24,25].

PA can be extracted from soybeans or eggs and is widely used as a shaping agent due to its muscle-enhancing effects [26,27]. Studies in athletes have shown that a daily PA intake of 750 mg, combined with strength training, significantly enhances muscle protein synthesis, increases lean body mass, and reduces fat content [28]. Within this process, the mTOR signaling pathway is significantly activated, suggesting that PA may be an effective intervention for fat metabolism. While the fat-reducing function of PA has been confirmed in athletes and the upstream regulator phosphatidic acid phosphatase (PAP) clearly plays a crucial role in lipid homeostasis [29,30], the specific function of PA has not been studied in models of obesity. Therefore, we established a model of obesity induced by a high-fat, high-sugar diet to validate the role of PA in obesity.

## 2. Materials and Methods

### 2.1. Reagents

PA (derived from soybean extract) was purchased from Zhenzhun Biotechnology Co., Ltd. (Shanghai, China), with the catalog number 02159016. Standard rodent chow was purchased from Qinglongshan Breeding Farm (Nanjing, Jiangsu, China), while the high-fat, high-sugar diet was custom-made by the farm according to our formulation (15% lard, 10% sugar, and 3% egg yolk powder added to the standard rodent chow). In the PA group, the corresponding weight percentage of PA was added to the high-fat, high-sugar diet.

### 2.2. Mice and HFD-Induced Obesity Model

Three-week-old, specific pathogen-free male C57BL/6 mice were purchased from the Model Animal Research Center of Nanjing University. All mice were acclimated to the surrounding environment for approximately 1 week prior to use. The animal housing conditions were maintained at SPF level, with a 12 h light–dark cycle, at room temperature (23 ± 2 °C), and a relative humidity of 40–70%. All animal experiments were carried out according to the NIH Guide for the Care and Use of Laboratory Animals [31] and were approved by the Experimental Animal Care and Use Committee of Southeast University.

For the HFD-induced obesity model, the mice were randomly divided into two groups. One group was fed standard rodent chow (*n* = 10), while the other group was fed a high-fat, high-sugar diet (*n* = 25). After 11 weeks, the group fed the high-fat, high-sugar diet had become obese model mice. In the 12th week, the obese model mice were further divided into three groups: the first group continued to be fed the high-fat, high-sugar diet (HFD, *n* = 7), the second group was fed a high-fat, high-sugar diet containing 0.25% PA (HFD + 0.25% PA, *n* = 8), and the third group was fed a high-fat, high-sugar diet containing 1% PA (HFD + 1% PA, *n* = 8). Normal mice continued to be fed standard rodent chow. The body weight of the mice was measured weekly throughout the study.

### 2.3. Detection of Biochemical Markers in Serum Samples

After fasting for 12 h, approximately 500 µL of blood was collected from mice via retro-orbital bleeding and placed in EP tubes. After standing for 1 h, the samples were centrifuged at 4 °C and 3000 rpm for 20 min. The supernatant was then collected and sent to the Department of Clinical Laboratory, Huai’an First People’s Hospital for analysis. The markers tested were triglycerides (TAGs), total cholesterol (CHDL), high-density lipoprotein (HDL-C), low-density lipoprotein (LDL-C), apolipoprotein A (APA), apolipoprotein B (APB), and lipoprotein a (LP(a)).

### 2.4. Immunohistochemistry of Tissue Samples

After fixation in 10% formalin for two days, mouse liver tissue was rinsed overnight in running water and subsequently embedded in paraffin the next day. The paraffin-embedded blocks were sectioned using a Leica microtome with a thickness of 4–6 µm. Following deparaffinization, dehydration, and antigen retrieval, the sections were stained with hematoxylin and eosin.

### 2.5. Oil Red Staining

Restore the frozen section to room temperature, fix it with tissue fixing solution (G1101, Servicebio, Wuhan, Hubei, China) for 15 min, wash with tap water, and dry. 6 parts of saturated oil red O dye solution (G1015, Servicebio, Wuhan, Hubei, China) and 4 parts of distilled water were fully mixed and homogenized, left to rest at 4 °C overnight, filtered once with qualitative filter paper the next day, placed at 4 °C for 24 h and filtered again to obtain oil red O working solution. Immerse the slices in the oil red dye solution for 8–10 min (cover to avoid light). Take out the slices, stay for 3 s, and then immerse in two cylinders of 60% isopropyl alcohol for differentiation, 3 s and 5 s, respectively. The slides were immersed in 2 tanks of pure water for 10 s each. Take out the slides, stay for 3 s, dip in hematoxylin G1004, Servicebio, Wuhan, Hubei, China) for 3–5 min and soak in 3 tanks of pure water for 5 s, 10 s and 30 s, respectively. Treat with the differentiation solution (G1039, Servicebio, Wuhan, Hubei, China) for 2–8 s, rinse in 2 tanks of distilled water for 10 s each, and blue in the blue solution (G1040, Servicebio, Wuhan, Hubei, China) for 1 s. The slides were gently immersed in 2 tanks of tap water for 5 s and 10 s each, and the staining effect was checked by microscopy. Seal the slides with glycerin gelatin (G1402, Servicebio, Wuhan, Hubei, China).

### 2.6. RNA Extraction, cDNA Synthesis, and Quantitative Real-Time PCR

Total RNA was extracted using the Trizol total RNA isolation reagent (Invitrogen™, Carlsbad, CA, USA) following the manufacturer’s instructions. Subsequently, cDNA was synthesized by using a High-Capacity cDNA Reverse Transcription Kit (Thermo Fisher Scientific, St. Louis, MO, USA) according to the manufacturer protocol. qRT-PCR was carried out using a SsoAdvanced™ Universal SYBR Green Supermix real-time PCR kit (Bio-Rad, Hercules, CA, USA). Primers were synthesized by Genscript (Nanjing, Jiangsu, China). The relative gene expression levels were determined using the 2−ΔΔCt method, while tubulin served as an internal control. All the qPCR primers are listed in Table 1.

### 2.7. RNA-Seq Analysis

Total RNA was extracted from 100 mg samples of liver tissue using Trizol total RNA isolation reagent (Invitrogen™). Enrichment, library construction, sequencing, and other RNA-related work were carried out by BGI Genomics in Shenzhen. The RNA-Seq data is available in Gene Expression Omnibus (GSE279177). The clean reads were mapped to the mm10 genome using TopHat and Cufflinks [32]. The statistical differences between groups were determined using calculations performed by Cuffdiff. The downstream analyses and plots were conducted in R 4.3.0(http://www.rproject.org/).

### 2.8. Statistical Analysis

Statistical analyses were performed using GraphPad Prism 7 software. The statistical methods are described in the Figure legends.

## 3. Results

### 3.1. Effect of PA on Obesity Induced by High-Fat, High-Sugar Diet

After 11 weeks of being fed a high-fat, high-sugar diet, the weight of the mice in the HFD group was significantly higher than that of the control group (Figure 1A,B). To validate the successful construction of the obesity model at the molecular level, we utilized q-PCR to detect the expression levels of marker genes related to lipid metabolism in the livers of mice fed normal diet (ND) and HFD. Among these genes, Fasn encodes fatty acid synthase, which catalyzes the synthesis of palmitate from acetyl-CoA and malonyl-CoA in the presence of NADPH. Ucp2 encodes mitochondrial uncoupling protein 2, which mediates the transport of protons from the intermembrane space to the inner mitochondrial membrane, reducing the potential difference across the inner and outer membranes without producing ATP and directly converting the potential energy into heat release. Both Agpat2 and Agpat3 encode 1-acylglycerol-3-phosphate O-acyltransferase, which acylates 1-acylglycerol-3-phosphate to form lysophosphatidic acid; in the mouse liver, Agpat2 is more highly expressed compared to Agpat3. The q-PCR results indicate that, compared to ND-fed mice, the expression levels of Fasn and Agpat3 significantly decreased in HFD-fed mice, while Ucp2 significantly increased and Agpat2 remained unchanged (Figure 1C), consistent with the previous literature [33]. Based on this, we conclude that our mouse obesity model was successfully established at the molecular level. On this basis, we divided the mice in the HFD group into three further groups and continued to feed them either the high-fat, high-sugar diet, the high-fat, high-sugar diet plus 0.25% PA, or the high-fat, high-sugar diet plus 1% PA. The results show that the 1% PA treatment significantly reversed the obesity induced by the high-fat, high-sugar diet. The 0.25% PA treatment, although effective, demonstrated a weaker effect compared to the pronounced effect observed in the 1% PA group (Figure 1D,E). Further whole-blood biochemical analysis showed that the upregulation of CHDL, TAG, HDL-C, and LDL-C caused by the high-fat, high-sugar diet did not change significantly after adding PA, but the level of LP(a) was significantly improved, which indicates that PA may improve HFD-induced obesity by adjusting the level of LP(a) (Figure 1F). In addition, histological staining showed that neither the high-fat, high-sugar diet nor PA treatment had a significant effect on liver steatosis or fibrosis (Appendix A), while the storage of lipid droplets in the 1%PA group was significantly lower than that in the HFD group (Appendix A).

### 3.2. PA Reversed HFD-Induced Obesity at the Transcriptomic Level

Based on our animal experiments, we found that PA intake could reverse obesity induced by a high-fat, high-sugar diet. Therefore, we intend to further investigate its mechanism of action. Although no apparent improvements in liver tissue immunohistochemistry were observed, there was a significant alteration in the storage of lipid droplets. Therefore, we focused more on changes at the molecular level. We took equal-weight samples of liver tissue from the various groups of mice and conducted RNA sequencing. Initially, we compared the ND group with the HFD group and found that the high-fat, high-sugar diet significantly altered gene expression in liver tissue, resulting in a total of 469 differentially expressed genes, of which 248 were upregulated and 221 were downregulated (Figure 2A). Through GO analysis, we discovered that the differentially expressed genes were significantly enriched in pathways related to stimulus response, immune system processes, regulation of metabolic processes, and fat metabolism, consistent with existing reports (Figure 2B). The genes enriched in the fat metabolism- and immune system-related pathways are highlighted in the MA plots (Appendix A).

We further analyzed the 469 differentially expressed genes in the two groups of mice receiving food containing PA. The results indicated that the addition of 0.25% PA did not significantly alter these 469 differentially expressed genes, but the addition of 1% PA led to a reversal in the expression of some genes (Figure 2C). Specifically, 59 genes were significantly increased in the HFD group and significantly decreased after the addition of 1% PA (genes downregulated by PA), while 48 genes showed a significant decrease in the HFD group and significantly increased after the addition of 1% PA (genes upregulated by PA). These 107 genes with significantly reversed expression levels are displayed in a separate heatmap (Figure 2D, Appendix A). GO analysis revealed that these reversed genes are concentrated in pathways responsible for the negative regulation of lipid metabolism (Figure 2E). Additionally, by comparing the genes that differed between the ND and HFD groups and between the ND and HFD + 1% PA groups, we found that the enrichment levels of pathways related to the immune system, fat metabolism regulation, and fat biosynthesis significantly decreased after the addition of 1% PA, with only the enrichment level of ketone metabolism significantly increasing (Figure 2E). These results indicate that PA reverses HFD-induced obesity at the transcriptional level.

### 3.3. PA Reverses Aberrant Expression of Key Genes in the PPAR Signaling Pathway

To further investigate the impact of PA on signaling pathways, we conducted a KEGG analysis on the genes with differential expression induced by the HFD (Figure 3A). We found that these differentially expressed genes were significantly enriched in multiple metabolic pathways, such as steroid hormone biosynthesis, retinol metabolism, and primary bile acid biosynthesis. Previous studies have also shown that a high-fat, high-sugar diet disrupts the balance of bile acids in the body, particularly intestinal bile acids, which can lead to the development of colon cancer. Additionally, it is noteworthy that the HFD-induced differentially expressed genes were significantly enriched in the PPAR signaling pathway (Figure 3B).

PPAR, a ligand-activated transcription factor nuclear receptor superfamily member, has three known subtypes: PPARɑ, PPARβ/δ, and PPARϒ. This pathway plays a crucial role in fat synthesis, lipid metabolism, insulin sensitivity, and inflammation by regulating the expression of related genes. PPARɑ and PPARϒ are mainly distributed in the liver and adipose tissue, while PPARβ/δ is abundantly expressed throughout the body with lower expression levels in the liver. PPAR can be activated by fatty acids and their derivatives. Aberrations in the PPAR signaling pathway are closely related to obesity and its complications [6], so it is not surprising that an HFD interferes with the expression of genes related to the PPAR signaling pathway.

In the PPAR signaling pathway, we observed that the HFD led to a decrease in the expression levels of Fabp5, Fabp7, Cyp7a1, and Cyp4a12b, while the expression levels of Fabp2 and Cyp4a14 increased (Figure 3B). Furthermore, we integrated the results from the HFD + 1% PA group into our analysis and found that the addition of 1% PA significantly reversed the expression levels of Fabp5 and Fabp7, two upstream molecules (Figure 3B, lower part), indicating PA’s regulatory effect on the PPAR signaling pathway.

To further validate our conclusions, we utilized q-PCR to confirm the expression of these genes. The results indicated that, in the liver tissue of mice in the HFD group, both the upstream genes of the PPAR signaling pathway (Fabp2, Fabp5, and Fabp7) and the downstream target genes (Cyp7a1, Cyp4a12b, and Cyp4a14) exhibited similar expression trends to those observed in the RNA-seq analysis (Figure 4). Moreover, the addition of 1% PA to the high-fat, high-sugar diet effectively reversed the transcriptional changes in Fabp5, Fabp7, and Cyp4a12b.

## 4. Discussion

Obesity leads to a range of health issues, and a high-fat, high-sugar diet is a key contributor to obesity. Our research findings indicate that the addition of PA to an HFD effectively inhibits weight gain in mice. Biochemical analyses revealed that the HFD significantly elevated serum levels of CHDL, TAG, HDL-C, LDL-C, and LP(a), suggesting an increased risk of cardiovascular diseases; however, when PA was added to the HFD, while there was no impact on the serum concentrations of CHDL, TAG, and LDL-C, HDL-C levels significantly increased and LP(a) levels decreased. LP(a) is a risk factor for atherosclerosis, implying that PA may have the potential to alleviate cardiovascular diseases by regulating LP(a) levels. Additionally, histological staining indicated that PA exerts a moderate effect by reducing lipid droplet storage, but it did not significantly improve liver steatosis.

Therefore, we shifted our focus to the transcriptome level, and further analysis revealed that mice fed a high-fat, high-sugar diet exhibited characteristics of obesity at the transcriptional level. Consuming the high-fat, high-sugar diet resulted in 469 differentially expressed genes, of which 248 were upregulated and 221 were downregulated. GO analysis of the differentially expressed genes showed that the genes affected by the HFD were involved in responses to stimuli, immune system processes, and lipid metabolism. Abnormalities in lipid metabolism and compromised immune systems are commonly observed in obese individuals. Upon the addition of PA to the high-fat, high-sugar diet, we observed a reversal in this differential expression, with GO analysis indicating that the reversed genes were primarily involved in pathways related to lipid metabolism regulation and immune system processes. Thus, we speculate that PA may reverse obesity progression in mice and alleviate inflammation by restoring lipid metabolism homeostasis and repairing damaged immune systems. By comparing the enrichment levels of genes induced by the high-fat, high-sugar diet and genes following the consumption of a high-fat, high-sugar diet supplemented with 1% PA in the same period, our hypothesis was further validated. Through KEGG enrichment analysis, we also found that the high-fat, high-sugar diet affected numerous pathways related to energy metabolism, including primary bile acid synthesis.

Furthermore, it is noteworthy that the genes that were differentially expressed in the mice fed a high-fat, high-sugar diet were also enriched in the PPAR signaling pathway. PPAR, as a nuclear receptor activated by fatty acids, plays a key role in regulating the expression of many genes involved in fat synthesis, lipid metabolism, insulin sensitivity, and inflammation. Within this signaling pathway, six genes exhibited changes in expression levels under the influence of the high-fat, high-sugar diet, especially Fabp5 and Fabp7, which were significantly downregulated under HFD conditions but showed reversed expression levels after the addition of PA. This suggests that PA may reverse the transcriptional changes induced by a high-fat, high-sugar diet through the regulation of the PPAR signaling pathway. We further validated this reversal effect at the transcriptional level through qPCR.

PA is widely used by athletes and fitness enthusiasts. In addition to promoting muscle growth, it also exhibits fat-reducing effects. Our research demonstrates that PA can not only aid fat reduction in the general population but also combat obesity induced by a high-fat, high-sugar diet. Derived from soy extracts, PA is a natural product with no adverse effects on the human body. Upon ingestion, it is metabolized by pancreatic phospholipase in the intestine to become lysophosphatidic acid and glycerol-3-phosphate. Therefore, PA holds promise as a potential safe and healthy additive with weight-reducing capabilities. However, as demonstrated, although PA can reverse the transcriptional changes induced by a high-fat, high-sugar diet, its molecular mechanisms are not yet fully understood and warrant further investigation.

## Figures and Tables

**Figure 1 genes-16-01112-f001:**
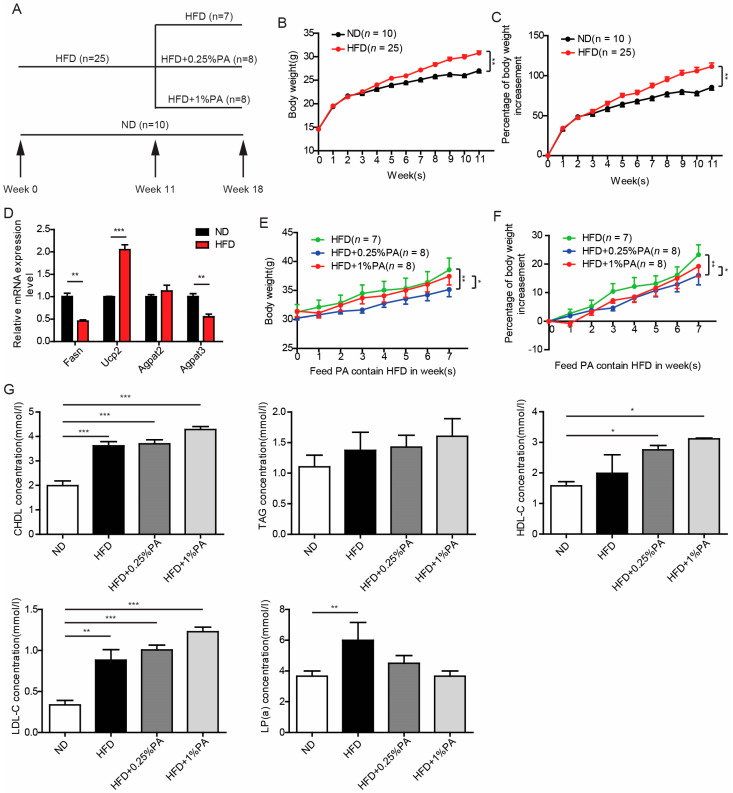
PA Inhibits Weight Gain Induced by HFD. (**A**) Schematic Diagram of Experimental Procedure. (**B**) Mouse weight gain curve. Mice were fed with a normal diet (ND) and a high-fat, high-sugar diet (HFD), and their weights were measured weekly. Mice fed with HFD showed significantly higher weight gain compared to those fed with ND. (**C**) Changes in mouse weight gain rate curve. The weekly weight gain rate was calculated, reflecting the growth rate of mouse weight relative to the initial weight. Mice fed with HFD exhibited a significantly higher weight gain rate compared to ND-fed mice. (**D**) q-PCR analysis of the expression levels of fat metabolism-related genes in ND and HFD-fed mice. In mice fed with HFD, the expression levels of Fasn and Agpat3 were significantly decreased, while the expression level of Ucp2 was significantly increased. (**E**) Following the pattern in Figure (**B**), mice were fed with an HFD containing different concentrations of PA (0.25%, 1%) starting from the 11th week, and their weight gain curves were monitored. Mice fed with HFD containing 0.25% and 1% PA showed lower weight gain compared to mice fed with HFD alone. (**F**) Weight rate curve of mice fed with HFD containing different concentrations of PA (0.25%, 1%). Mice fed with HFD containing 0.25% and 1% PA exhibited lower weight gain rates compared to those fed with HFD alone. Statistical methods for Figures (**B**,**C**,**E**,**F**): two-way ANOVA, Mean ± SEM, * *p* < 0.05, ** *p* < 0.01. Statistical method for Figure (**D**): *t*-test, Mean ± SEM, ** *p* < 0.01, *** *p* < 0.001. (**G**) Concentrations of total cholesterol (CHDL), triglycerides (TAG), high-density lipoprotein (HDL-C), low-density lipoprotein (LDL-C), and lipoprotein a [LP(a)] in mouse serum under different dietary conditions. Statistical analysis was performed using one-way ANOVA, with post hoc analysis using Tukey’s test, Mean ± SEM, * *p* < 0.05, ** *p* < 0.01, *** *p* < 0.001.

**Figure 2 genes-16-01112-f002:**
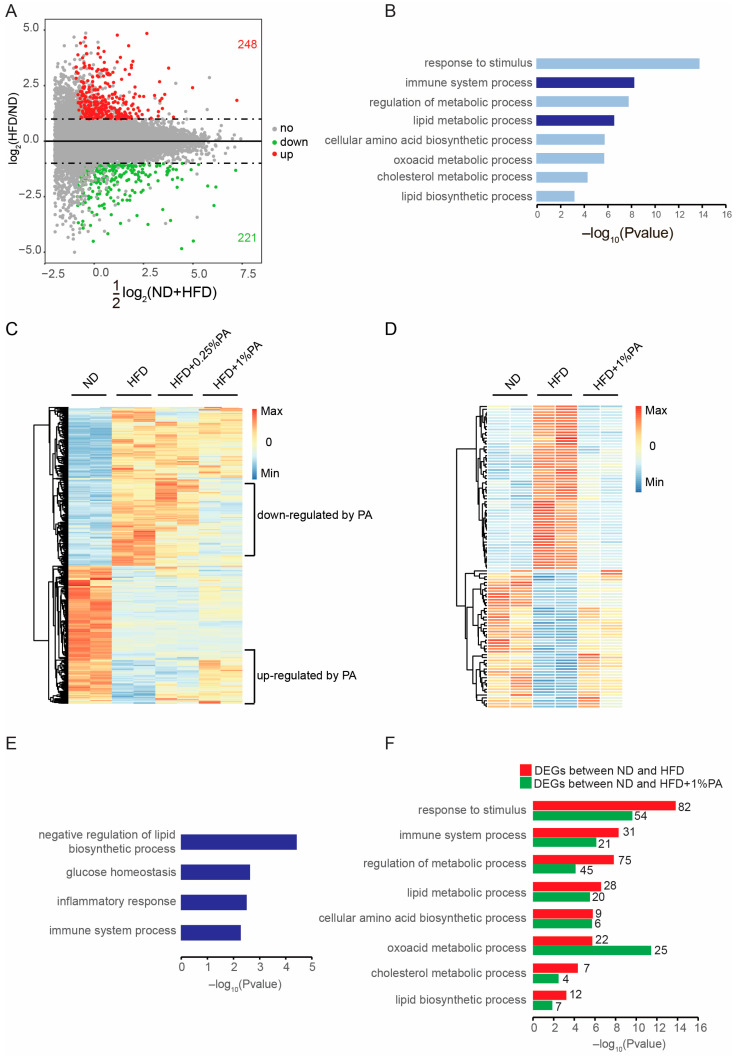
High-fat, high-sugar diet can impact gene expression changes in lipid metabolism and immune pathways at the transcriptional level, while PA is capable of partially reversing the transcriptional alterations induced by high-fat, high-sugar diet. (**A**) Changes in the liver transcriptome of mice from the ND group and the HFD group. The MA plot shows 248 upregulated genes (log2FC > 1, in red) and 221 downregulated genes (log2FC < 1, in green). (**B**) GO analysis reveals that differentially expressed genes between the ND and HFD groups are significantly enriched in immune and metabolic-related biological processes, with deep blue highlighting immune system processes and lipid metabolism processes. (**C**) We clustered and visualized the changes in 469 differentially expressed genes among the mouse groups fed with different diets in Figure 3 using a heatmap. We observed that the significantly upregulated genes in HFD could be divided into two major groups: one group showed no change after the addition of PA, while the other group significantly decreased after the addition of PA. Similarly, the significantly downregulated genes in HFD could also be divided into two major groups: one group showed no change after the addition of PA, while the other group significantly increased after the addition of PA. Both classes of genes regulated by PA reversed the effects of HFD under the influence of PA. (**D**) Statistical analysis was performed on the genes regulated by PA in panel (**C**) using Cuffdiff. The results indicated that 59 genes were significantly downregulated after the addition of PA, while 48 genes were significantly upregulated after the addition of PA. We displayed these 107 genes in a heatmap. (**E**) Results of the GO analysis on these 107 genes in panel (**D**). (**F**) Gene Ontology (GO) analysis was performed on the differentially expressed genes between the ND group and the HFD group, as well as between the ND group and the HFD group supplemented with 1% PA. The comparison revealed that the addition of 1% PA can reverse the transcriptional changes in pathways related to lipid metabolism, immune response, and other pathways.

**Figure 3 genes-16-01112-f003:**
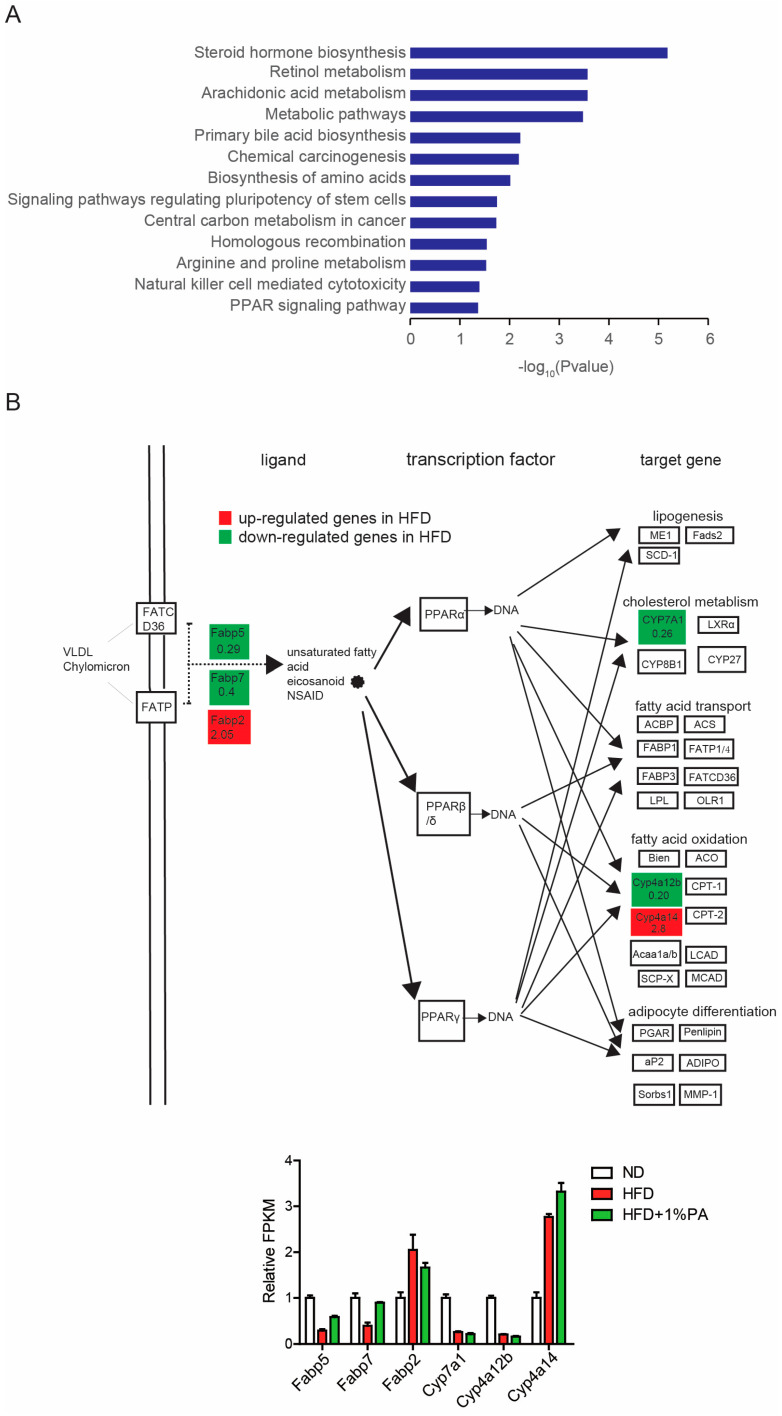
PA reverses the changes in the PPAR pathway induced by a high-fat, high-sugar diet. (**A**) Results of KEGG pathway enrichment for differentially expressed genes between the ND and HFD groups. (**B**) The presentation of the PPAR signaling pathway, highlighting the differentially expressed genes between the ND and HFD groups, with the data after PA supplementation shown below.

**Figure 4 genes-16-01112-f004:**
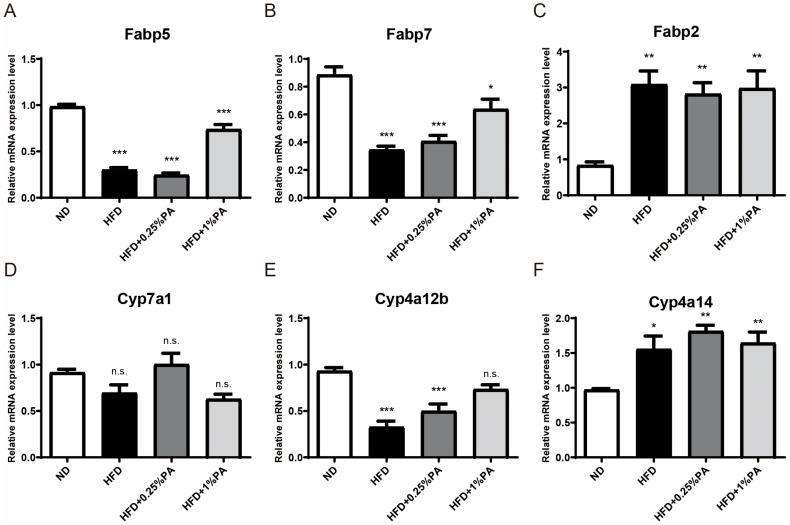
PA indeed reverses the gene expression of the PPAR pathway in the liver tissues. We assessed the expression levels of Fabp5 (**A**), Fabp7 (**B**), Fabp2 (**C**), Cyp7a1 (**D**), Cyp4a12b (**E**), and Cyp4a14 (**F**) in the liver tissues of mice fed with different diets using qPCR. Statistical analysis was performed using one-way ANOVA, with post hoc analysis using Tukey’s test, Mean ± SEM, n.s. no significance, * *p* < 0.05, ** *p* < 0.01, *** *p* < 0.001.

**Table 1 genes-16-01112-t001:** PCR primers used in this study.

Primer Names	Primer Sequences	Expected Amplicon Size	GENCODE Gene Names
Tubulin-F	AGCTTTGGCGGGGGAACTGG	167 bp	Tuba1a
Tubulin-R	AGGGTGGTGTGGGTGGTGAGGAT		
Fasn-F	CTGGCCCCGGAGTCGCTTGAGTAT	101 bp	Fasn
Fasn-R	AAGGCGCACAGGGACCGAGTAATG		
Ucp2-F	TCGCCTCCCCTGTTGATGTGGTC	112 bp	Ucp2
Ucp2-R	GCGCGGGGTCCCTCCTTCC		
Agpat2-F	CATCAACCGCCAGCAAGCCAGAAC	199 bp	Agpat2
Agpat2-R	AGACGAGTACACCACGGGGATGAT		
Agpat3-F	CCGCTTGGCCTACTCGCTCTG	189 bp	Agpat3
Agpat3-R	ACGCCAAACCGCTCGCACAT		
Fabp5-F	GGAAGTGGCGCCTGATGGA	142 bp	Fabp5
Fabp5-R	GTTTTGACCGTGATGTTGTTGC		
Fabp7-F	AAGTGGGATGGCAAAGAAACAAAT	101 bp	Fabp7
Fabp7-R	TCATAACAGCGAACAGCAACGATA		
Fabp2-F	CGGCACGTGGAAAGTAGACC	189 bp	Fabp2
Fabp2-R	ACACCGAGCTCAAACACAACATCA		
Cyp7a1-F	GCCTTCTGCTACCGAGTGATGTTT	188 bp	Cyp7a1
Cyp7a1-R	CGGGCTTTATGTGCGGTCTTG		
Cyp4a12b-F	CCCCCTGTACCAAGTGTGAG	183 bp	Cyp4a12b
Cyp4a12b-R	GCTGTGCCGGGAAGACC		
Cyp4a14-F	TGCAGAAGGCCAGGAAGAAGAGA	180 bp	Cyp4a14
Cyp4a14-R	GGGTGGCCAGAGCATAGAAAATC		

## Data Availability

RNA-Seq data is available in Gene Expression Omnibus (GSE279177).

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
