# Peer review of "Phosphatidic Acid Reverses Obesity Induced by a High-Fat, High-Sugar Diet at the Transcriptional Level"

_genes, 2025, doi:10.3390/genes16091112_

Round 1

Reviewer 1 Report

Comments and Suggestions for Authors

Review Comments:

The manuscript by Hao Xie al., entitled “ Phosphatidic acid reverses obesity induced by a high-fat diet at 2 the transcriptional level”. The author investigates the transcriptional changes associated with PA treatment in a HFD- induced obese animal model by performing Transcriptome analysis (next-generation sequencing) of liver tissue. Hence, the overall conclusions of the manuscript are likely correct, but there are a couple of Major comments that need to be addressed. Some of the individual issues are listed below.

Comments:

Introduction: Background and references are insufficient; the author should discuss major lipid-regulating genes and the role of PA in membrane and storage lipid homeostasis.

Materials and methods:

  • The author should include an experimental flow chart in the “Mice and HFD-induced fat model” section or in Fig.1.
  • “Immunohistochemistry and tissue microarray analysis” – this section only discusses tissue sectioning; the methodology for tissue microarray analysis is unclear and should be provided in more detail.

Results and discussion:

  • Line 120: sentence is not clear; the author should rewrite it with more precise details for better clarity, and a rationale for the selected gene should be stated here (i.e, Fasn, Ucp2, Agat2 …)
  • The author should check the typo on page 3, line 121:  “makers” should be “markers”. Please carefully proofread the typos throughout the manuscript.
  • On page 6. The author only shows the BMI graph in Fig.1. Why were representative images of the ND, HFD, HFD+0.25% PA, and HFD+0.25% PA mice not included?
  • The manuscript would benefit from presenting the rationale of the study prior to describing the results.
  • Lines 138 and 139: The statement “Based on the previous animal experiments” requires an appropriate reference.
  • Supplementary file is missing?
  • The manuscript lacks continuity throughout, making it difficult to follow the flow of ideas.
  • The author should also show storage lipids (Lipid droplets) in liver tissue using Oil Red O staining in all experimental groups.

Reviewer 2 Report

Comments and Suggestions for Authors

Review Comments on genes-3856765

Journal: Genes 

Manuscript ID: genes-3856765

Title: Phosphatidic acid reverses obesity induced by a high-fat diet at the transcriptional level

Authors: Hao Xie *, Qian Cheng, Xingyi Tian, Yanlin Liao

Molecular Genetics and Genomics

Major comments: 

  • Phosphatidic acid (PA) is one of the simplest diacylglycerol phospholipids, serving as a crucial precursor for the synthesis of triglycerides and other complex phospholipids. Additionally, PA is an important intermediate product in the process of fat digestion and absorption. Studies have shown that daily intake of 750mg of PA by athletes, combined with strength training, significantly enhances muscle protein synthesis, increases lean body mass, and reduces fat content. In this process, the mTOR signaling pathway is significantly activated, suggesting that PA may be an effective intervention for fat metabolism. While the fat-reducing function of PA has been confirmed in athletes and the upstream regulator phosphatidic acid phosphatase (PAP) clearly plays a crucial role in lipid homeostasis, the specific function of PA in obesity models has not been studied.
  • Based on these thoughts, in the present study, authors analyzed whether addition of PA to food can reverse obesity induced by a high-fat diet or not.
  • Authors stated that the addition of PA to food can reverse obesity induced by a high-fat diet. They also stated that transcriptomic analysis results indicate that the reversal effect of PA exists at the molecular level. Authors stated further that further analysis suggests that PA may regulate fat metabolism by reversing the PPAR signaling pathway.
  • These findings seemed very interesting observations. However, due to following apparent defects, this manuscript cannot be acceptable in Genes.

  • Authors stated that there was a big difference between 0.25% PA vs. 1% PS. They described that “the high-fat and high-fructose diet plus 1% PA treatment group can significantly reverse the obesity induced by the high-fat and high-fructose diet, but 0.25% PA has no obvious effect (Figure1D, E)” (page 3, lines 126-128) However, based on my own inspection on Figure 1D, E, I could not support the authors description at all. How can authors conclude so?
  • What is the basis of the difference in PA concentration in the diet (0.25% PA vs. 1% PA), since there was no description why these values were chosen in the present study. How authors chose 0.25% PA vs. 1% PA concentrations?
  • In the Materials and Methods section, authors described that “the first group continued be fed the high-fat diet (HFD, n=7), the second group was fed a high-fat diet containing 0.25% PA (HFD + 0.25% PA, n=8), and the third group was fed a high-fat diet containing 1% PA (HFD + 1% PA, n=8)” (page 2, lines 76-79) and no description on a high-fat and high-fructose diet (or a high-fat and high-sugar diet) at all. However, in Results section, it is described as follows “They were given a high-fat and high-fructose diet, a high-fat and high-fructose diet plus 0.25% PA, and a high-fat and high-fructose diet plus 1% PA” (page 3, lines 124-125). Which is correct? It is very confusing.
  • Authors described in the Results section that “Based on the previous animal experiments, we found that the intake of PA can reverse obesity induced by a high-fat high-fructose diet” (page 4, lines 138-139). The results of these previous animal experiments are apparently very important for understanding of the present study. If these studies had already published, their reference(s) are absolutely required. If not, these results should be included in the present manuscript.
  • Over all, both “high-fat and high-fructose diet” and “high-fat diet (HFD)” were used in the manuscript. Both conditions were used or only “high-fat and high-fructose diet” was used in the present study?

Minor comments:

  • “ND group” was used without prior definition. (page 4, line 143)
  • In each Figure captions, (a), (b), (c), (d),… might be better to change (A), (B), (C), (D),…, respectively, for clarification, since panel numbers in each Figure are in capital letter.

Round 2

Reviewer 1 Report

Comments and Suggestions for Authors

In this revised version of the manuscript, the author has addressed the reviewers’ concerns. In my opinion, the manuscript is now suitable for publication.